# Metabolites from Aerial Parts of *Glycyrrhiza foetida* as Modulators of Targets Related to Metabolic Syndrome

**DOI:** 10.3390/biom14040467

**Published:** 2024-04-11

**Authors:** Hekmat B. Al-Hmadi, Elena Serino, Arianna Pastore, Giuseppina Chianese, Saoussen Hammami, Mariano Stornaiuolo, Orazio Taglialatela-Scafati

**Affiliations:** 1Department of Chemistry, College of Medicine, Al-Muthanna University, Samawah 66001, Iraq; hekmat.alhmadi@mu.edu.iq; 2Laboratory of Environmental Chemistry and Clean Processes (LR21ES04), Faculty of Sciences of Monastir, Monastir University, Monastir 5000, Tunisia; h_saoussen@yahoo.fr; 3Department of Pharmacy, School of Medicine and Surgery, University of Naples Federico II, Via D. Montesano 49, 80131 Napoli, Italy; elena.serino@unina.it (E.S.); arianna.pastore@unina.it (A.P.); g.chianese@unina.it (G.C.)

**Keywords:** *Glycyrrhiza foetida*, metabolic syndrome, amorfrutins, mitochondrial activity, GLUT

## Abstract

A detailed phytochemical investigation has been carried out on the aerial parts of *G. foetida* leading to the isolation of 29 pure compounds, mainly belonging to the amorfrutin and polyphenol classes. Among them, the new amorfrutin N (**5**) and exiguaflavone L (**21**) were isolated and their structures elucidated by means of HR-ESIMS and NMR. All the isolated compounds were investigated for modulation of mitochondrial activity and stimulation of glucose uptake via GLUT transporters, two metabolic processes involved in intracellular glucose homeostasis, which, therefore, correlate with the incidence of metabolic syndrome. These experiments revealed that amorfrutins were active on both targets, with amorfrutin M (**17**) and decarboxyamorfrutin A (**2**) emerging as mitochondrial stimulators, and amorfrutin 2 (**12**) as a glucose uptake promoter. However, members of the rich chalcone/flavonoid fraction also proved to contribute to this activity.

## 1. Introduction

Metabolic syndrome is an alarmingly spreading pathological condition that includes a cluster of risk factors specific for cardiovascular disease, such as abdominal obesity, hypertension, impaired fasting glucose, high triglyceride levels, and low HDL cholesterol levels [1]. Generally, these conditions have a multifactorial origin, but the main triggering factors are high-sugar and high-fat diets associated with a sedentary lifestyle. Consequently, the first line of interventions to treat metabolic syndrome consist of a change in the nutritional and lifestyle behaviors of the affected people. If not adequately curbed, metabolic syndrome can evolve into severe pathologies that can often co-occur, such as obesity, dyslipidemia, cardiovascular diseases, and diabetes mellitus. It has been estimated that diabetes causes over one million deaths per year, and this rate is expected to increase in the future. 

Many natural products, especially those belonging to the polyphenol class, are currently in the spotlight for their range of benefits on endpoints related to the onset of metabolic syndrome. Thus, several plant-derived extracts or dietary supplements are in the market to prevent and treat metabolic syndrome [2]. In this context, one of the most interesting and promising class of polyphenolic metabolites is represented by amorfrutins, biogenetically hybrid natural products, isolated from different Fabaceae (Leguminosae) species and based on a polyketide-derived 2,4-dihydroxybenzoic acid core decorated with a prenyl unit para-oriented to a substituent whose nature mirrors one of the ketide starters. Amorfrutins have been recognized as potent and selective agonists of PPARγ receptors [3], interacting on a site that is different from those of the synthetic thiazolidinediones, and thus, putatively associated with lower side effects [3]. PPARγ agonism of amorfrutins is associated with a potent insulin-sensitizer effect with a beneficial effect on diabetes symptoms. It has been postulated that the concomitant activation of different PPARs, including PPARα, involved in fatty acid oxidation, can lead to greater efficiency in simultaneously targeting different aspects of metabolic syndrome [4]. 

In a recent investigation of the aerial parts of a Tunisian specimen of *Glycyrrhiza foetida* (Fabaceae), aimed at characterizing the amorfrutin composition of this plant, we discovered several phenethyl- and alkyl-type amorfrutins that were evaluated for agonist activity on PPARα and PPARγ nuclear receptors [5]. As expected, amorfrutin A was a potent agonist of PPARγ, but it was also discovered that amorfrutin H is a selective PPARα agonist, and amorfrutin E is a dual PPARα/γ agonist [5]. 

These encouraging results, coupled with the traditional reported use of *G. foetida* to treat hepatic and metabolic disorders [6], prompted us to carry out a more comprehensive NMR-based phytochemical screening on this plant. Indeed, besides the somewhat rare amorfrutins, present only in another couple of species of the *Glycyrrhiza* genus (*G. acanthocarpa* and *G. lepidota*) [7,8], additional metabolites are likely to contribute, with different mechanisms, to the positive effects of this plant on metabolic syndrome progression [9].

A detailed purification of the total extract of *G. foetida* aerial parts led to the isolation of 29 compounds, including 16 previously characterized amorfrutins (**1**–**4** and **6**–**17**) and a new one named amorfrutin N (**5**) (Figure 1). In addition, six flavanone derivatives (**18**–**23**), including the unprecedented exiguaflavone L (**21**), were obtained. Moreover, the isoflavonoid biochanin A (**24**), the chalcone phlorizin (**25**), the acetophenone myrciaphenone A (**26**), the acylphloroglucinol 6-dihydroxyphenyl-1-butanone-2-β-D-O-glucopyranoside (**27**), benzyl-β-D-glucopyranoside (**28**), and the triterpene oleanolic acid (**29**) were also isolated (Figure 2). All the pure compounds obtained in sufficient amounts were tested in three different pipelines to identify the active constituents and their possible roles in treating dysfunctional pathways commonly involved in metabolic syndrome. In particular, mitochondrial activity, hypoglycemic effects, and lipid homeostasis modulation were evaluated, disclosing some interesting bioactivities on these targets.

## 2. Materials and Methods

### 2.1. General Experimental Procedure

Optical rotations (CHCl_3_ and MeOH) were measured at 589 nm on a P2000 (JASCO Europe s.r.l., Cremella, Italy) polarimeter. ^1^H (600 and 700 MHz) and ^13^C (150 and 175 MHz) NMR spectra were measured on a Bruker spectrometer. Chemical shifts are referenced to the residual solvent signals (CDCl_3_: δ_H_ 7.26, δ_C_ 77.0; CD_3_OD: δ_H_ 3.31, δ_C_ 49.3). Homonuclear ^1^H connectivity was determined by COSY experiments. Through-space ^1^H connectivities were evidenced by using a NOESY experiment with a mixing time of 300 ms. One-bond heteronuclear ^1^H−^13^C connectivities were determined by the HSQC experiment; two- and three-bond ^1^H−^13^C connectivities were determined by gradient-HMBC experiments optimized for a ^2,3^*J* of 8 Hz. HRESIMS experiments were performed on an LTQ-Orbitrap mass spectrometer equipped with an ESI interface and Excalibur data system. MPLC-DAD separations were performed on an Interchim instrument, puriFlash XS 520 Plus (Sepachrom s.r.l., Rho, Italy). HPLC-RI separations were performed on Knauer instruments, using apparatus and columns as described in [5]. RP-HPLC-UV−vis separation was performed on an Agilent instrument, using a 1260 Quat pump VL system, equipped with a 1260 VWD VL UV−vis detector using a Luna 10 μm C18 100 Å 250 mm × 10 mm column and a Rheodyne injector. Thin-layer chromatography (TLC) was performed as described in [5].

### 2.2. Plant Material 

*G. foetida* specimens were collected in the Fernana region (Northwestern Tunisia) in July 2019. The plant was identified by Dr. Ridha El Mokni, Department of Pharmaceutical Sciences “A”, Laboratory of Botany, Cryptogamy and Plant Biology, Faculty of Pharmacy of Monastir, Tunisia, where a voucher specimen has been deposited with a code GF 121.

### 2.3. Extraction 

*G. foetida* aerial parts were air-dried in the shade, ground, and powdered to obtain 600 g of material that was extracted as reported in ref. [5], affording 80 g of a brown crude extract. 

### 2.4. LC-UV-MS^2^ Analysis 

LC-UV-MS^2^ analysis was carried out on an LTQ-XL Ion Trap mass spectrometer (Thermo Fisher, Waltham, MA, USA) equipped with an Ultimate 3000 HPLC. The MS and MS^n^ spectra, in negative mode, were recorded using the same parameters as described in [5].

### 2.5. Purification of Metabolites from the G. foetida Extract

The crude extract was chromatographed on an open column (90 cm length, 4 cm diameter) packed with silica (230–400 mesh). Compounds were eluted with a gradient starting from *n*-hexane followed by *n*-hexane/EtOAc in the following ratios: 8:2, 7:3, 1:1, 4:6, 2:8, 1:9, and then EtOAc, and continuing with EtOAc/MeOH 9:1, 7:3, 1:1, 3: 7, 1:9, and finally MeOH. Equally sized fractions of 200 mL each were collected in clean glass flasks, monitored by TLC, and similar ones were pooled to obtain fractions E1 to E18.

Based on their similar TLC profiles, fractions E2 to E5 were combined (2.0 g) and purified via MPLC-DAD (silica cartridge 50 Å 25 µ—size 25, 25 g)—column volume (CV) 30 mL/min. The mobile phase was a mixture of A: hexane, B: EtOAc, and C: MeOH, with the gradient method: A-B 9:1 for CV 0–5; from A-B 9:1 to A-B 1:1 for CV 5–40; isocratic A-B 1:1 for CV 40–45; from A-B 1:1 to B for CV 45–50; isocratic B for CV 50–52; isocratic C for 53–55 CV; the flow rate was 25.0 mL/min. UV detection wavelength = 275 nm. This separation afforded 16 fractions that were further purified by HPLC. Fractions 3, 4, 5, and 7 were purified on a HPLC-RI detector equipped with a Rheodyne injector, on a Luna 5 µm Silica (**2**) 100 Å 250 mm × 4.6 mm column, with a flow rate of 1.0 mL/min, using hexane and EtOAc as the mobile phase. Fraction 3 (68.4 mg) was separated by using *n*-hexane/EtOAc 95:5 as the mobile phase, to afford decarboxyamorfrutin 2 (**13**) (R_t_ = 6.5 min, 18.4 mg) and amorfrutin M (**17**) (R_t_ = 9.0 min, 1.6 mg). Fraction 4 (67.2 mg) afforded decarboxyamorfrutin A (**2**) (R_t_ = 15.0 min, 4.7 mg), amorfrutin 2 (**12**) (R_t_ = 16.0 min, 8.6 mg), the new amorfrutin N (**5**) (R_t_ = 21.3 min, 0.8 mg), and decarboxydemethoxyamorfrutin C (**10**) (R_t_ = 24.8 min, 3.0 mg), while fraction 5 (64.3 mg), purified using *n*-hexane/EtOAc 9:1, led to the isolation of amorfrutin C (**11**) (R_t_ = 8.6 min, 2.0 mg) and amorfrutin A (**1**) (R_t_ = 9.5 min, 15.4 mg). Fraction 7 (50.3 mg) was separated using *n*-hexane/EtOAc 85:15 as the eluent to yield amorfrutin H (**8**) (R_t_ = 19.7 min, 2.0 mg), hiravanone (**23**) (R_t_ = 21.0 min, 6.8 mg), and lonchocarpol A (**22**) (R_t_ = 22.0 min, 16.9 mg). Fraction 8 (19.9 mg) was purified in the same conditions, leading to the isolation of oleanolic acid (**29**) (R_t_ = 20.0 min, 1.7 mg). Fraction 9 (75.5 mg) was purified using *n*-hexane/EtOAc 8:2 to yield sakuranetin (**18**) (R_t_ = 17.8 min, 1.6 mg) and biochanin A (**24**) (R_t_ = 18.9 min, 1.0 mg). Fraction 10 (15.7 mg) was purified using *n*-hexane/EtOAc 70:30, affording sophoraflavanone B (**19**) (R_t_ = 19.5 min, 1.4 mg) and exiguaflavanone K (**20**) (R_t_ = 21.2 min, 1.6 mg), while fraction 11 (31.3 mg) was purified in the same conditions as fraction 10, yielding the new exiguaflavone L (**21**) (R_t_ = 16.5 min, 2.7 mg). Fraction E7 (40.7 mg) was purified using *n*-hexane/EtOAc 7:3 to afford amorfrutin J (**14**) (R_t_ = 4.2 min, 2.1 mg), amorfrutin L (**16**) (R_t_ = 8.6 min, 1.9 mg), and amorfrutin I (**9**) (R_t_ = 17.5 min, 2.0 mg). Fraction E9 was purified by RP-HPLC-RI using MeOH/H_2_O 7:3 + 0.1% of formic acid on a Kinetex 264 C18 100 Å 100 mm × 4.6 mm column, with a flow rate of 1.0 mL/min, to afford amorfrutin F (**6**) (R_t_ = 8.2 min, 2.4 mg), amorfrutin K (**15**) (R_t_ = 9.5 min, 1.9 mg), and amorfrutin 3 (**3**) (R_t_ = 14.9 min, 1.7 mg). Fraction E13 (429.1 mg) was first purified by MPLC-DAD on a Silica C18 Cartridge 50 Å 25 µ—size 40 (55 g)—column volume (CV) 30 mL/min, eluting with *n*-hexane/EtOAc 70:30, with a flow rate of 20 mL/min. This purification led to the isolation on eight fractions. Fractions 13_2 and 13_4 were purified by HPLC-RI on a Luna C18 (2) 250 mm × 4.6 mm column, 5 µm, by using 65:35 H_2_O + 0.1% of formic acid/methanol as the mobile phase. Fraction E13_2 (15.7 mg) led to the isolation of myrciaphenone A (**26**) (R_t_ = 6.7 min, 1.6 mg) and benzyl-β-D-glucopyranoside (**28**) (R_t_ = 8.3 min, 2.1 mg), while fraction E13_4 afforded 4,6-dihydroxyphenyl-1-butanone-2-β-D-O-glucopyranoside (**27**) (R_t_ = 10.6 min, 5.6 mg) and phlorizin (**25**) (R_t_ = 14.2 min, 6.8 mg).

#### 2.5.1. Amorfrtuin N (**5**)

Colorless amorphous solid [α]_D_ 0 (c 0.16, CHCl_3_). HRESI-MS, *m*/*z* 313.1814 [M + H]^+^ (calc. for C_20_H_25_O_3_ *m*/*z* 313.1804). ^1^H and ^13^CNMR: see Table 2.

#### 2.5.2. Exiguaflavone L (**21**)

Colorless amorphous solid [α]_D_—4.2 (c 0.19, CHCl_3_). HRESI-MS, *m*/*z* 403.1390 [M + H]^+^ (calc. for C_21_H_23_O_8_ *m*/*z* 403.1393). ^1^H and ^13^CNMR: see Table 3.

### 2.6. Mitochondrial Activity

Mitochondrial activity was measured in HuH7 and C2C12 cells using a MitoTracker^®^ Red CMXRos probe (Thermo Fisher Scientific, Waltham, MA, USA). The probe was diluted in DMEM to a final a dye working solution of 100 nM. HuH7 and C2C12 cells were washed twice in PBS before being cultivated in the presence of the probe for 60 min in a cell incubator set to 37 °C and 5% CO_2_. Upon incubation, cells were extensively washed in DMEM and once in PBS, and then fixed in 3.7% formaldehyde for 30 min. MitoTracker fluorescence was measured using a Perkin Elmer Envision 2105 Multiplate Reader (Perkin Elmer, Waltham, MA, USA) with the following parameters: λexcitation at 579 nm and λemission at 599 nm. Cells were then permeabilized in 0.1% Triton X-100 in PBS and stained with the nuclear dye DAPI. DAPI fluorescence was measured using a λexcitation at 351 nm and λemission at 450 nm. In order to normalize mitochondrial activity to the total number of cells, Mitotraker fluorescence was normalized to DAPI fluorescence.

### 2.7. NBDG Glucose Uptake Assay

C2C12 cells were seeded (5 × 10^3^/well) on a 96-well black bottom microtiter plate (Perkin Elmer, Waltham, MA, USA) in a final volume of 100 μL/well of growth media. When the cells achieved 80–90% confluency, the culture medium was carefully removed and replaced with 100 μL of HBSS containing 100 μM 2-deoxyglucose, 0.4 g/L Bovine Serum Albumin, and 1.27 mM CaCl_2_ (without any growth factors or FBS). When indicated, test molecules were added to the wells at the indicated concentration in DMSO and cells incubated for 1 h at 37 °C. Upon incubation, the culture medium was further supplemented with 64 μM of the fluorescent glucose tracer 2-NBDG. After 45 min of incubation with the fluorescent probe, plates were washed twice in PBS. Uptake of 2-NDBG was determined using a Perkin Elmer Envision 2105 Multiplate Reader (Perkin Elmer) with the following parameters: λexcitation at 471 nm, λemission at 529 nm, and monochromator cut-off at 360 nm. Following the determination of 2-NDBG, cells were fixed in 3.7% paraformaldehyde for 30 min, permeabilized in 0.1% Triton X-100 in PBS, and stained with the nuclear dye DAPI (100 μM). The parameters of λexcitation at 351 nm and λemission at 450 nm were used to measure the fluorescence of DAPI. Normalized glucose uptake is indicated as the ratio between intracellular 2-NDBG fluorescence and DAPI fluorescence.

### 2.8. RNA Purification and mRNA Quantification by Real-Time RT-PCR 

Total RNAs were purified from Huh7 cell lines with an RNeasy^®^Plus Mini Kit (QIAGEN, Hilden, Germany) according to the manufacturer’s protocol. A mass of 0.7 µg of purified RNAs were converted to cDNAs with a High-Capacity cDNA Reverse Transcription Kit (Applied biosystems, Thermo Fisher Scientific, Waltham, MA, USA) as recommended by the manufacturer. The cDNAs were amplified by real-time RT-PCR in a Step One Plus Real-Time PCR System (Applied biosystems, Thermo Fisher Scientific) with the fluorescent double-stranded DNA-binding dye SYBR™ Green PCR Master mix (Applied biosystems, Thermo Fisher Scientific) according to the manufacturer’s protocol. Specific primers for each gene were designed to work under the same cycling conditions (95 °C for 10 min followed by 40 cycles at 95 °C for 15 s, 52 °C for 30 s, and 72 °C for 30 s), thereby generating products of comparable size (about 100–400 bp for each amplification). Primer combinations were positioned whenever possible to span an exon–exon junction and the RNA was digested with DNAse to avoid genomic DNA interference. All samples were run in triplicate and the results, expressed as N-fold differences in target gene expression, were determined as follows: =2^(−(ΔCT[target] − ΔCT[control]). Primer sequences for each gene analyzed were designed as follows: CYP17A1: FW 5′-GAGCTTGAGGCACGAGAAC-3′ RV 5′-TGGAATGGTGTTTGCTTGCG-3′; PGC1α: FW 5′-ATTGCCCTCATTTGATGCGC-3′ RV: 5′-TAGCTGAGTGTTGGCTGGTG-3′; GAPDH: FW 5′-GCACCACCAACTGCTTA-3′ RV:5′-AGTAGAGGCAGGGATGAT-3′.

### 2.9. GC/MS Analysis 

HuH7 cells (2 × 10^6^) were scraped in ice-cold water and centrifuged at 10,000× *g* for 5 min at 4 °C. Membrane pellets were dried and dissolved in 1 mL of ice-cold dichloromethane. Insoluble material was removed by centrifugation at high speed for 10 min at 4 °C. The supernatants were dried and resuspended in acetonitrile. Samples were solubilized in pyridine (50 μL) and derivatized with 25 μL of *N*,*O*-bis(trimethylsilyl(TMS)trifluoroacetamide (BSTFA) with a reaction time of 90 min. A volume of 1 μL was injected, with a split ratio of 1:10. GC-MS analyses were carried out on a Shimadzu GCMS 2010 plus (Kyoto, Japan) with the following parameters: Injection temperature 280 °C, ramp 0–1.00 min 100 °C, 1.00–6.00 min 100–320 °C, hold for 2.33 min. Column flow 1.10 mL/min. Linear velocity 39 cm/s. Helium gas was used. Ion source temperature 200 °C, interface 320 °C, solvent cut 5.9 min, scan 35–600 *m*/*z*. Detector voltage 0.1 kV. Separation was performed on an Agilent (Santa Clara, CA, USA.) SIL-8, 30 m × 0.25 mm column, 0.25 μm. Cholesterol-TMS (*m*/*z*: 458, 368; 329; 129; 73; 75), palmitic acid-TMS (*m*/*z*: 313; 117; 73), and myristic acid-TMS (*m*/*z*: 285; 145; 117; 73) were identified by comparing their retention time and fragmentation to library standards.

## 3. Results and Discussion

### 3.1. Extraction, Isolation, and Structural Elucidation 

Air-dried, ground, and powdered aerial parts of *G. foetida* were exhaustively extracted with MeOH/CH_2_Cl_2_ 1:1 and the obtained extract was subjected to a preliminary LC-UV-MS^2^ analysis to carry out a first dereplication step (Figure 3). As expected, the extract was rich in amorfrutins, which were easily detected thanks to their fragmentation pattern and their diagnostic UV spectra with λ_max_ 220, 270, and 305 nm, in agreement with the literature [5]. However, the LC/MS chromatogram also showed peaks with different UV profiles, suggesting the presence of polyphenols belonging to other families that could be inferred by their typical UV absorption features. In fact, some of them, like flavanones and chalcones, share a similar fragmentation pattern but significantly differ in their UV profiles [10]. The acquired UV spectra were compatible with the flavanone skeleton, with λ_max_ 220, 295, and 345 nm (Table 1). Then, the most abundant amorfrutins, including amorfrutin A, the main component of the mixture (R_t_ 11.77 min), amorfrutin 2 (R_t_ 12.49 min), amorfrutin 3 (R_t_ 3.02 min), and amorfrutin C (R_t_ 16.0 min), were annotated based on their MS/MS profile. The flavanones lonchocarpol A (R_t_ 10.47 min) and hiravanone (R_t_ 10.77 min), and the isoflavone biochanin A (R_t_ 3.17) were also annotated and confirmed with NMR after isolation. 

Taking advantage of the information coming from the LC/MS profile, we carried out a first chromatographic purification on silica gel, eluting with a gradient of increasing polarity from *n*-hexane to MeOH, and obtained 18 fractions. These were extensively purified by HPLC and led to the isolation of 29 pure compounds, whose structures were deduced by application of NMR and HRMS. Comparison of their spectral data with those reported in the literature was instrumental to the identification of the known compounds. The isolated metabolites included 17 amorfrutins, 11 of which belonging to the phenetyl series and were identified as amorfrutin A (**1**), decarboxyamorfrutin A (**2**), amorfrutin 3 (**3**), amorfrutin E (**4**), amorfrutin F (**6**), amorfrutin G (**7**), amorfrutin H (**8**), amorfrutin I (**9**), decarboxydemethoxyamorfrutin C (**10**), amorfrutin C (**11**), and the new amorfrutin N (**5**). In addition, six amorfrutins belonging to the pentyl series were obtained and identified as amorfrutin 2 (**12**), decarboxyamorfrutin 2 (**13**), amorfrutin J (**14**), amorfrutin K (**15**), amorfrutin L (**16**), and amorfrutin M (**17**). All the known amorfrutins were identified by comparison of experimental NMR data with those acquired in our previous work [5]. 

The new amorfrutin N (**5**), molecular formula C_20_H_24_O_3_ deduced by HR-ESIMS, was isolated in minute amounts and its structure inferred by spectral data (see Appendix A) and comparison with known amorfrutins. In particular, its ^1^H NMR spectrum (Table 2) showed signals of two singlets in the aromatic region (δ_H_ 6.26 and 6.49) and one -OMe (δ_H_ 3.74), suggesting that **5** is a monomethoxylated and decarboxylated amorfrutin. In addition, typical signals of the phenylethyl moiety and side chain signals overlapping with those of amorfrutin 3 (**3**) were detected. The structure of **5** was secured by COSY, HSQC, and HMBC experiments and all the ^1^H and ^13^C NMR signals were assigned as reported in Table 1. Thus, amorfrutin N (**5**) is a new member of this class of secondary metabolites that can be viewed as a decarboxylated amorfrutin 3. Since **5** showed no optical rotation, it is racemic at C-2′′. 

Ten additional polyphenols and the triterpene oleanolic acid (**29**) [11] were also isolated from extract fractionation. In particular, six flavanone derivatives were obtained and identified as sakuranetin (**18**) [12], sophoraflavanone B (**19**) [13], exiguaflavanone K (**20**) [14], the unprecedented exiguaflavone L (**21**), lonchocarpol A (**22**) [15], and shiravanone (**23**) [16]. In addition, the isoflavonoid biochanin A (**24**) [17], the chalcone phlorizin (**25**) [18], the acetophenone myrciaphenone A (**26**) [19], 6-dihydroxyphenyl-1-butanone-2-β-D-O-glucopyranoside (**27**) [20], and benzyl-β-D-glucopyranoside (**28**) [21] were also obtained. The known compounds were identified by comparison of their spectral data with those reported in the literature [12,13,14,15,16,17,18,19,20,21].

Exiguaflavone L (**21**) was obtained as a pale yellow amorphous solid with the molecular formula C_21_H_22_O_8_, as deduced by the positive HR-ESIMS peak at *m/z* 425.1211 [M + Na]^+^ (calculated for C_21_H_22_O_8_Na 425.1212). The ^1^H NMR of **21** (Table 3), analyzed with the help of a COSY spectrum, showed typical signals of a prenylated flavanone of the exiguaflavone family, including the ring A singlet at δ_H_ 6.07, the CH_2_-CH spin system of ring B (δ_H_ 5.32, 3.09 and 2.79), and three almost completely overlapped methine signals around δ_H_ 6.93. The prenylated side chain included a -CH_2_CH- moiety (δ_H_ 3.08, 2.79, 4.48), a *sp^2^* methylene at δ_H_ 5.07, and a methyl singlet at δ_H_ 1.87. All the proton signals were associated with those of the directly linked carbon atoms thanks to the correlations of the HSQC spectrum. The carbon skeleton of exiguaflavone L (**21**) was built up with the aid of the HMBC spectrum, with the crucial correlations of H-1 with C-1′, C-2′, and C-6′, of H-6 with C-5, C-10, C-7, and C-8, and of H-1″ with C-3, C-7, C-8, and C-9. The placement of the methoxy group was secured by the HMBC cross-peak of the singlet at δ_H_ 3.93 with C-3′. Finally, a hydroperoxy group, in agreement with the molecular formula, was placed at C-2″ on the basis of the marked downfield shift of the carbon atom (δ_C_ 88.8). The minute amounts isolated prevented determination of the absolute configuration at C-2″. 

### 3.2. Biological Evaluation on Targets Related to Metabolic Syndrome

In order to identify secondary metabolites responsible for the beneficial effects on metabolic syndrome traditionally attributed to the plant, we used in vitro cultures of muscle and liver cells to measure the effect of isolated *G. foetida* metabolites on (i) mitochondrial activity, (ii) glucose uptake, and (iii) lipid homeostasis. 

#### 3.2.1. Effect on Mitochondrial Activity

In the first pipeline, we investigated the effect of *G. foetida* metabolites on the mitochondrial activity of C2C12 myoblasts grown in vitro. Mitochondrial activity was measured using the selective probe MitoTracker CMXRos [22]. The probe accumulates in the mitochondrial intermembrane space and emits fluorescence with an intensity well correlated to cellular mitochondrial activity. 

C2C12 cells were cultured in the presence of *G. foetida* pure metabolites. As shown in Figure 4, most of the amorfrutins tested (**1**–**3**, **6**, **8**, **9**–**17**) boosted mitochondrial activity in C2C12, although with different potencies. Amorfrutin M (**17**) and decarboxyamorfrutin A (**2**) clearly presented the highest potencies. Sophoraflavanone B (**19**) also showed a good activity. Confocal microscopy confirmed mitochondrial stimulation of the amorfrutin-containing fraction (AF) obtained from the total extract (E2 and E3) (Figure 4).

To verify the mitochondrial-stimulating activity of amorfrutins on a different cell line, we tested *G. foetida* pure metabolites on human hepatoma HuH7 cells (Figure 5). Similar to myoblasts, amorfrutins also exerted mitochondrial stimulatory activity on the hepatoma cell line. Amorfrutin A (**1**), its decarboxylated analogue **2**, amorfrutin C (**11**), its decarboxylated and demethoxylated analogue **10**, and amorfrutin M (**17**), presented the highest potencies. 

To investigate at the transcriptional level the effect of *G. foetida* metabolites on mitochondrial metabolism, mRNA was extracted from HuH7 cells treated with the amorfrutin-rich fraction AF. Analyzed by qPCR, mRNA levels of PGC1-α (PPARγ coactivator 1alpha), a key player in lipid metabolism and mitochondrial activity [23], were upregulated upon incubation with the AF (Figure 6). On the contrary, the enzyme involved in cholesterol conversion into bile acids, CYP7A1, was downregulated in treated cells, suggesting that anabolic reactions involving cholesterol were hampered in cells treated with amorfrutins. A reduction in lipid content (fatty acids and cholesterol) was confirmed by GC-MS. Lipids were extracted from HuH7 cells treated for 48 h with the AF. GC-MS analysis revealed decreased intracellular levels of cholesterol, palmitic acid, and myristic acid, probably as consequence of augmented β-oxidation promoted by PPARγ (Figure 6).

#### 3.2.2. Glucose Uptake Modulatory Activity

In a second pipeline, *G. foetida* metabolites were tested for their capacity to promote glucose uptake via glucose transporters (GLUTs). Dysmetabolic disorders, which are frequently characterized by insulin resistance and hyperglycemia, have impaired control over the handling of circulating glucose levels. GLUTs are essential for removing glucose from the bloodstream, and many of them are controlled by insulin receptors and react to insulin stimulation via PI3K signaling. Insulin stimulation can increase the number of GLUT transporters (mostly GLUT4) on the plasma membrane of muscle cells, boost glycolysis, and ultimately enhance the activity of passive transporters (GLUT1). The effect of *G. foetida* metabolites on glucose uptake was measured by monitoring the uptake of NBDG [2-(7-nitro-2,1,3-benzoxadiazol-4-yl)-D-glucosamine], a fluorescent derivative of deoxyglucose covalently bound to the fluorescent probe nitro blue tetrazolium (NBT) [24]. An intracellular increase in fluorescence upon NBDG uptake indicates activation of the GLUT transporters. As shown in Figure 7, most of the amorfrutins exhibited glucose uptake stimulatory activity, with amorfrutin 2 (**12**) presenting the highest potency (Figure 7). 

## 4. Conclusions

In summary, our detailed investigation of the aerial parts of *G. foetida* led to the isolation of 29 pure compounds mainly belonging to the amorfrutin class and to other polyphenol classes. These compounds included amorfrutin N (**5**) and exiguaflavone L (**21**), new members of the amorfrutins and prenylated flavanones, respectively. All the isolated compounds were investigated for modulation of mitochondrial activity and stimulation of glucose uptake via GLUT transporters, two endpoints related to glucose homeostasis and, thus, to metabolic syndrome. Our results revealed that amorfrutins were active on both targets, with amorfrutin M (**17**) and decarboxyamorfrutin A (**2**) emerging as mitochondrial stimulators, and amorfrutin 2 (**12**) as a glucose uptake promoter. However, the rich polyphenol components of this extract clearly contribute to this activity, and sophoraflavanone B (**19**) and 6-dihydroxyphenyl-1-butanone-2-β-D-O-glucopyranoside (**27**) were active in mitochondrial and GLUT stimulation, respectively. Moreover, it is worth mentioning that the extract also contains the glycosylated dihydrochalcone phlorizin (**25**), a sodium-glucose transport (SGLT) protein inhibitor in the nephron with a significant effect on inhibiting reabsorption of sugar in the kidney and a consequent lowering effect on plasma glucose [25]. The activity of phlorizin served as inspiration for the development of the synthetic class of SGLT2 inhibitors collectively named glifozins [26].

In conclusion, since the *G. foetida* total extract contains several biomolecules acting on different targets potentially involved in the management of metabolic syndrome symptoms, its traditional use is fully supported by our investigation, which, hopefully, will inspire further analysis to support a commercial use of this herbal product. 

## Figures and Tables

**Figure 1 biomolecules-14-00467-f001:**
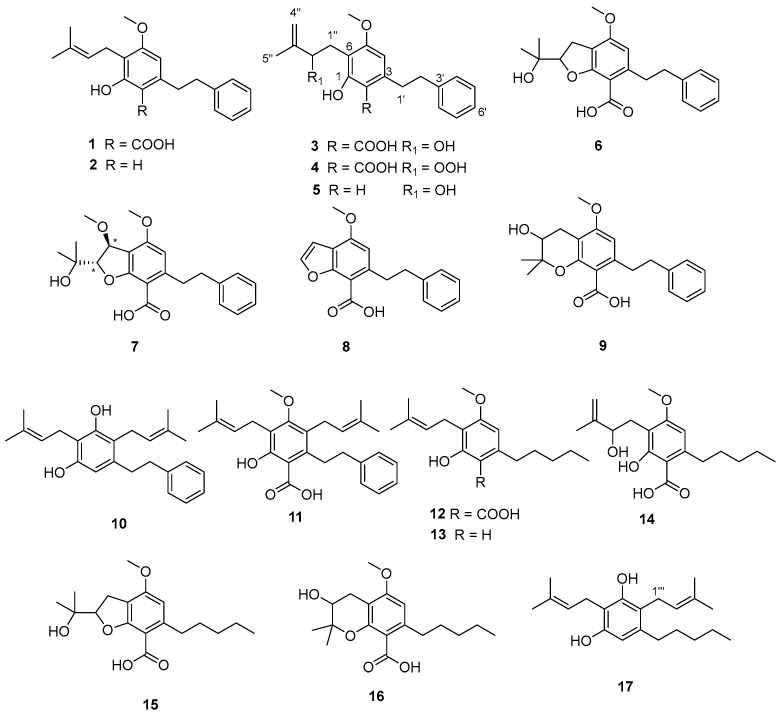
Amorfrutins isolated in this study from aerial parts of *G. foetida.*

**Figure 2 biomolecules-14-00467-f002:**
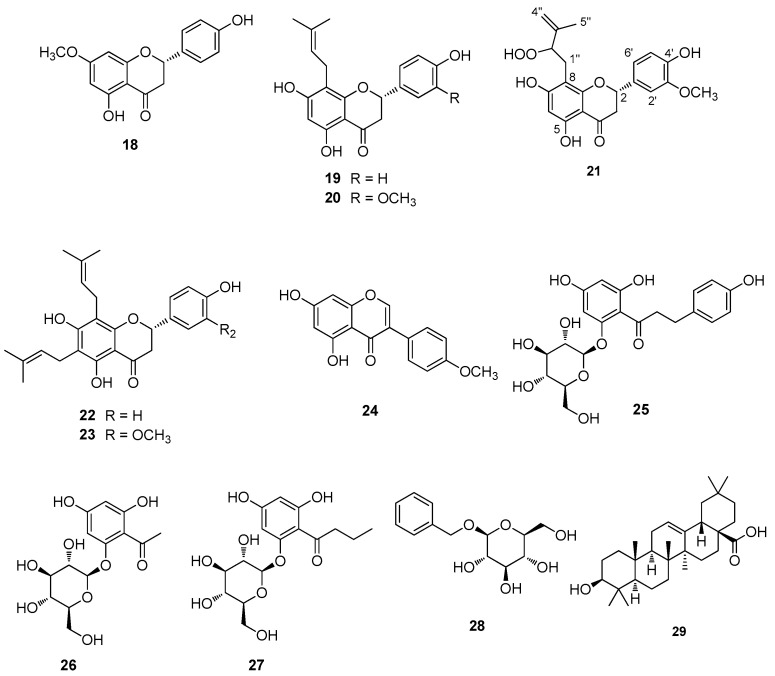
Chalcones, flavonoids, and other metabolites isolated from *G. foetida*.

**Figure 3 biomolecules-14-00467-f003:**
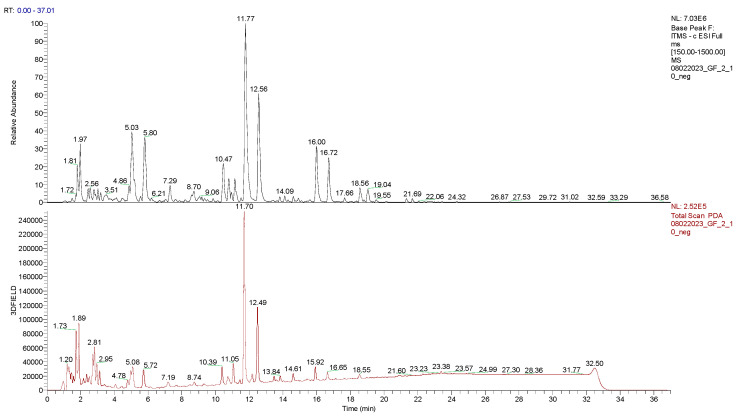
LC-MS base ion peak chromatogram (**up**) and LC-UV (**bottom**) chromatogram of the extract obtained from the aerial parts of *G. foetida*.

**Figure 4 biomolecules-14-00467-f004:**
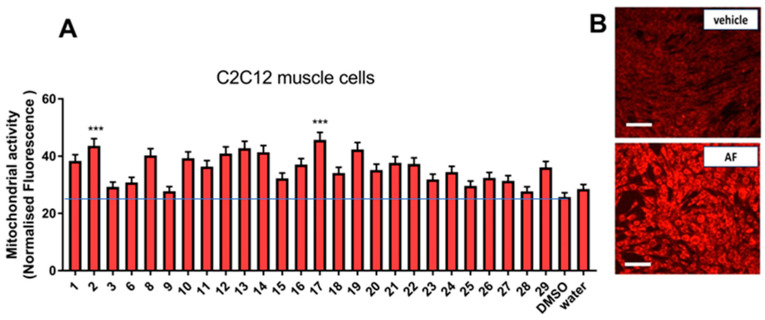
Effect of *G. foetida* pure metabolites on mitochondrial metabolism in C2C12 cells. (**A**) Normalized Mito-tracker Red CMX-ROS fluorescence measured upon incubation of C2C12 cells with the indicated *G. foetida* metabolites (3 μM) for 48 h (values are reported as mean ± SEM and are representative of three independent experiments; *** *p* < 0.001, all the others *p* < 0.05); (**B**) confocal microscopy of C2C12 cells incubated with pooled amorfrutins (Amorfrutin Fraction (AF), 10 mg/L) or with an equivalent amount of vehicle for 48 h and stained with Mito-tracker Red CMX-ROS. Magnification bar = 100 μm. Blue line marks the level of control.

**Figure 5 biomolecules-14-00467-f005:**
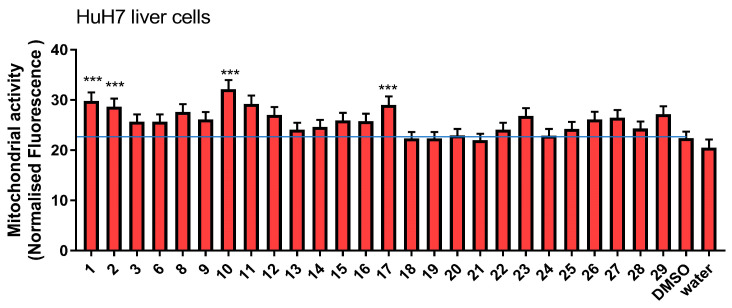
Effect of *G. foetida* pure metabolites on mitochondrial metabolism in HuH7 liver cells. Normalized Mito-tracker Red CMX-ROS fluorescence measured upon incubation of HuH7 cells with the indicated *G. foetida* metabolites (3 μM) for 48 h. (Values are reported as mean ± SEM and are representative of three independent experiments; *** *p* < 0.001, all the others *p* < 0.05). Blue line marks the level of control.

**Figure 6 biomolecules-14-00467-f006:**
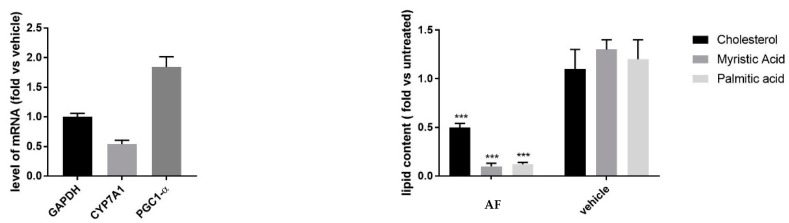
(**Left panel**) Intracellular level of PGC-1α and CYP7A1 mRNA in HuH7 cells treated with *G. foetida* amorfrutin-enriched fraction AF (10 mg/L) for 48 h (values are reported as mean ± s.d. and are representative of three independent experiments). GAPDH (glyceraldehyde-3-phosphate dehydrogenase); (**right panel**) intracellular levels of cholesterol, Palmitic Acid, and Myristic Acid in HuH7 cells treated with *G. foetida* fraction AF (10 mg/L) for 48 h (values are reported as mean ± s.d. and are representative of three independent GC-MS experiments, *** *p* < 0.001).

**Figure 7 biomolecules-14-00467-f007:**
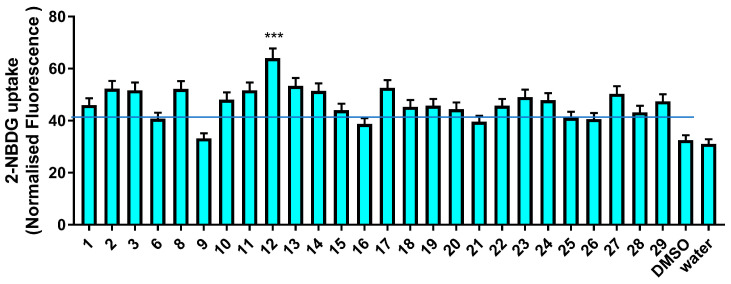
Normalized 2-NBDG fluorescence measured upon incubation of C2C12 cells with the indicated *G. foetida* metabolites (3 μM) for 48 h; the dotted red line indicates the basal glucose uptake activity of the cells (values are reported as mean ± SEM and are representative of three independent experiments; *** *p* < 0.001, all the others *p* < 0.05). Blue line marks the level of control.

**Table 1 biomolecules-14-00467-t001:** Annotation of the major peaks of the *G. foetida* extract.

Assignment	*R*_t_ (min)	Precursor ion (*m*/*z*)	Fragments	UV (λ_max_ in nm)
Amorfrutin 3	3.02	355.30	311.28; 293.24;	220.00; 260.00; 300.00
Biochanin A	3.17	283.28	268.06; 265.20; 240.29	220.00; 260.00;
Exiguaflavanone K	9.06	369.33	339.12; 325.18; 293.23 284.17;	220.00; 265.00; 290.00; 375.00
Lonchocarpol A	10.47	407.33	398.22; 301.11; 313.21; 287.24; 261.22; 243.18	220.00; 295.00; 345.00
Hiravanone	10.77	437.43	422.15; 313.20; 301.18; 287.11; 261.25; 243.23	220.00; 295.00; 345.00 (220; 290; 335; 375; 425)
Amorfrutin A	11.77	339.28	295.15; 225.19	225.00; 270.00; 305.00; 375.00; 425.00
Amorfrutin 2	12.56	305.24	290.24; 275.25; 261.18; 247.22	220.00; 270.00; 305.00; 375.00; 420.00
Amorfrutin C	16.00	407.33	363.30; 348.25; 331.27; 305.12; 285.29	220.00; 315.00; 375.00; 410.00

**Table 2 biomolecules-14-00467-t002:** ^1^H NMR (600 MHz) and ^13^C NMR (150 MHz) of compound **5**.

Position	δ_H_, Mult., *J* in Hz	δ_C_, Mult.
1		155.8, C
2	6.26, s	103.5, CH
3		142.7, C
4	6.49, s	110.3, CH
5		158.2, C
6		112.6, C
1′	2.85, m	37.9, CH_2_
2′	2.91, m	37.8, CH_2_
3′		141.5, C
4′	7.20, overlapped	128.4, CH
5′	7.28, overlapped	128.6, CH
6′	7.20, overlapped	125.9, CH
7′	7.28, overlapped	128.6, CH
8′	7.20, overlapped	128.4, CH
1a″ 1b″	3.10, dd, 14.6, 2.3 2.78, dd, 14.6, 8.0	30.8, CH_2_
2″	4.29, dd, 8.0, 2.3	76.8, CH
3″		146.9, C
4a″ 4b″	4.86, s 4.99, s	110.3, CH_2_
5″	1.84, s	18.1, CH_3_
-OMe	3.74, s	55.5, CH_3_

**Table 3 biomolecules-14-00467-t003:** ^1^H NMR (600 MHz) and ^13^C NMR (150 MHz) of exiguaflavone L (**21**).

Position	δ_H_, Mult., *J* in Hz	δ_C_, Mult.
2	5.32, m	79.2, CH
3a 3b	3.09, m 2.79, m	43.4, CH_2_
4		196.0, C
5		161.9, C
6	6.07, s	96.4, CH
7		163.7, C
8		105.3, C
9		163.5, C
10		102.8, C
1′		135.9, C
2′	6.96, overlapped	108.8, CH
3′		146.4, C
4′		146.3, C
5′	6.94, overlapped	114.4, CH
6′	6.93, overlapped	119.6, CH
OMe	3.93, s	55.9, CH_3_
1**″**a 1**″**b	3.08, m 2.79, m	23.8, CH_2_
2**″**	4.48, m	88.8, CH
3**″**		143.4, C
4**″**	5.07, s	113.5, CH_2_
5**″**	1.87, s	18.9, CH_3_

## Data Availability

Data are contained within the article and Appendix A.

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
