# Peer review of "Metabolites from Aerial Parts of Glycyrrhiza foetida as Modulators of Targets Related to Metabolic Syndrome"

_biomolecules, 2024, doi:10.3390/biom14040467_

Round 1

Reviewer 1 Report

Comments and Suggestions for Authors

This manuscript described the purification of 29 compounds, including amorfrutins, chalcones, flavonoids and other metabolites, from G. foetida aerial parts. A new amorfrutin and a new flavonoid were reported. The authors further tested these compounds’ biological activities using mitochondrial activity and glucose uptake assays. 

1.     While the NMR data for the two novel compounds were provided, data supporting the annotation of the known 27 compounds were insufficient. Since authentic standards for these compounds were not mentioned, authors should provide either MS or NMR spectra for these compounds along with the known spectra from literature. 

2.     In the methods section describing the purification of metabolites, key parameters were missing. Specifically, insufficient information was given regarding the silica column’s size, particle size, etc (line 134). Was it an open-column chromatography? The gradient for this chromatography step was not mentioned either. It was unclear why “fraction E2 to E5 were combined” (line 138) or why “Fractions 3, 4, 5 and 7 were purified” (line 145). How did the authors assess the purity of the isolated compounds? What was the solvent system for fraction E13 (Line 168)? Why chose this fraction to further purify?

3.     Why the authors chose to analyze the aerial parts of the plant? Why not the roots?

4.     In the results, please explain briefly the structural characteristics of amorfrutin, chalcones, and flavonoids, respectively.

5.     Do the 29 compounds have any chemical identifiers such as CAS number, InChI, etc? If so, please provide.

6.     Line 269-271, “Thus, the most abundant amorfrutins, including…., were annotated”. How could UV spectra alone annotate these compounds?

7.     For the biological assays, statistical analysis was inadequate to support some of the conclusions. Specifically, samples with p<0.001 were noted with “***”, but whether data obtained from other samples were statistically significant was unknown. Thus, authors are suggested to mark data with p<0.05 to support conclusions including: “most of the amorfrutins tested boosted mitochondrial activity” (line 347), “Sophoraflavanone B (19) also showed a good activity” (line 349), “However, the rich polyphenol components of this extract clearly contribute to this activity…” (line 430-433).

8.     Why 48 hours were used for the biological assays?

9.     Fig 6, no gene expression data for PPARl was presented although it was mentioned in the text. Also, please provide the full name for the reference gene GAPDH. Finally, please explain the “***” in the legend. 

10.  Line 374, please add a reference for PGC1-a

11.  Line 404, please provide the full name for NBDG.

12.  Please correct grammatic errors in the text.

Comments on the Quality of English Language

Need to correct grammatical errors in the text

Author Response

1.     While the NMR data for the two novel compounds were provided, data supporting the annotation of the known 27 compounds were insufficient. Since authentic standards for these compounds were not mentioned, authors should provide either MS or NMR spectra for these compounds along with the known spectra from literature. 
Answer: The NMR spectra have been provided for the new compounds. On the other hand, we isolated also 16 known amorfrutins whose NMR were exactly identical to those reported in our previous work (J. Nat. Prod. 2023, 86, 2435-2447). This has been better specified in the manuscript (“All the known amorfrutins were identified by comparison of experimental NMR data with those acquired in our previous work [5].”). Similarly, we isolated 10 known flavanones/polyphenols and also in this case we identified them by comparing NMR data with those reported in the literature. We have better specified this in the text (” The known compounds were identified by comparison of their spectral data with those reported in the literature [12-21].).

2.     In the methods section describing the purification of metabolites, key parameters were missing. Specifically, insufficient information was given regarding the silica column’s size, particle size, etc (line 134). Was it an open-column chromatography? The gradient for this chromatography step was not mentioned either. It was unclear why “fraction E2 to E5 were combined” (line 138)
Answer: all the information requested by the reviewer have now been provided

 Why “Fractions 3, 4, 5 and 7 were purified” (line 145). 
Answer: Actually, all the fractions were purified. Fractions 3, 4, 5 and 7 have been combined in that sentence because they were purified with the same system.

How did the authors assess the purity of the isolated compounds? What was the solvent system for fraction E13 (Line 168)? Why chose this fraction to further purify?
Answer: The purity was assessed by their 1H NMR spectrum. The information on Fraction E13 has been added. We purified almost all the fractions, and the selection was based on preliminary 1H NMR spectra of the crude fractions. 

3.     Why the authors chose to analyze the aerial parts of the plant? Why not the roots?
Answer: As clarified in the Introduction (lines 52-58), this is a follow-up of a previous study on the aerial parts of this plant, which were found to be rich of amorfrutins. This is the reason for continuing the work on this part of the plant. 

4.     In the results, please explain briefly the structural characteristics of amorfrutin, chalcones, and flavonoids, respectively.
Answer: As requested, we have added structural features of amorfrutins in the Introduction (lines 44-46). We respectfully believe that the features of chalcones and flavonoids are well known to the readers.  

5.     Do the 29 compounds have any chemical identifiers such as CAS number, InChI, etc? If so, please provide.
Answer: The known compounds are indexed and have a CAS number. However, these identification numbers are commonly not included in natural products papers (also in this journal), because these compounds can be easily searched by using their trivial names.

6.     Line 269-271, “Thus, the most abundant amorfrutins, including…., were annotated”. How could UV spectra alone annotate these compounds?
Answer: The annotation of amorfrutin was not based on UV spectra but on MS/MS profile. UV was used to distinguish between isomeric chalcones and flavanones and between these and amorfrutins. We have more clearly specified this in the text (lines 273 and 275). 

7.     For the biological assays, statistical analysis was inadequate to support some of the conclusions. Specifically, samples with p<0.001 were noted with “***”, but whether data obtained from other samples were statistically significant was unknown. Thus, authors are suggested to mark data with p<0.05 to support conclusions including: “most of the amorfrutins tested boosted mitochondrial activity” (line 347), “Sophoraflavanone B (19) also showed a good activity” (line 349), “However, the rich polyphenol components of this extract clearly contribute to this activity…” (line 430-433).
Answer: Actually, we verified that the most active compounds had a significance of p<0.001, but all the other data had a p<0.05. As suggested, this has been indicated in the legends of the three figures. 

8.     Why 48 hours were used for the biological assays?
Answer: We have followed the same protocol adopted in our previous studies. 48 hours is the optimal time to see the maximal biological response. 

9.     Fig 6, no gene expression data for PPARl was presented although it was mentioned in the text. Also, please provide the full name for the reference gene GAPDH. Finally, please explain the “***” in the legend. 
Answer: It was a mistake. We have specified that PGC1-α is the PPARγ coactivator 1alpha. We have specified the full name of GADPH in the legend. Also in the legend we have explained the meaning of  ***. 

10.  Line 374, please add a reference for PGC1-a
Answer: As requested, a reference has been added for PGC1-a (ref. n 23)

11.  Line 404, please provide the full name for NBDG.
Answer: The name has been added

12.  Please correct grammatic errors in the text.
Answer: The text has been checked for grammatic errors and corrections made. 

Reviewer 2 Report

Comments and Suggestions for Authors

The paper is dedicated to the in-depth identification of bioactive metabolites from the Glycyrrhiza foetida plant. Preliminary research indicated the presence of some promising activity in its extracts, so such an investigation was reasonable.
The topic is relevant to the field, but the significance of the obtained results is moderate. First, from the provided measurements, it is not clear what could be the resultant effect on the metabolic syndrome. Second, no comparison to other molecules proposed to treat this syndrome was carried out.
The employed methods are fine; the combined NMR-MS-MS procedure should be enough for the identification of the novel flavonoids' structures.
The conclusions are supported by the data.

Specific comments:
Figure 4, 5, 7: It would be easier to understand the figure, if there were a horizontal reference line at the level of the control.
Figure 6: control values from the untreated cells are required

Comments on the Quality of English Language

Grammar and readability check using some online service could be useful to correct strange constructs, which sometimes occur in the text.

Author Response

Figure 4, 5, 7: It would be easier to understand the figure, if there were a horizontal reference line at the level of the control.

Answer: We thank the reviewer for the good suggestion. We have drawn a horizontal line at the level of control in Figures 4, 5, and 7.

Figure 6: control values from the untreated cells are required.

Answer: On the right panel of figure 6 there are explicit levels of vehicle (untreated cells). On the left panel, the scale is relative to the vehicle (untreated cells), thus CYP7A1 is almost half of the vehicle, while PGC1-alpha is almost the double of the vehicle.

Reviewer 3 Report

Comments and Suggestions for Authors

Current report has been investigated the aerial parts of Glycyrrhiza foetida leading to the isolation of 29 pure compounds. Please conduct the concerns below.

1.      Target on the aerial parts of a Tunisian specimen of Glycyrrhiza foetida (Fabaceae) needs to introduce in detail.

2.      Extraction must follow the previous report(s) with reference(s).

3.      How to prepare the purified compound for bioassay in cells?

4.      Assay of mitochondrial activity needs the reference(s) to support. Same as that of glucose uptake in C2C12 cells.

5.      Data in figures belonged to small size in triplicate only.

6.      In vivo data were ignored. Why?

7.      Dose-dependent change by each compound is extremely required.

8.      One table indicated the amount of each compound from this plant is helpful.

9.      Compound belonged to active in mitochondrial and GLUT stimulation must supported by reliable results.

10.  Phlorizin (25) is known as inhibitor of sodium-glucose transport protein without selectivity for type-1 or type-2. It seems not the same as conclusion.

11.  Criteria of metabolic syndrome needs to make clear. Please check the established reference(s).

12.  Effective dose of each compound was not conducted in clear. Novelty of current report was unknown.

Comments on the Quality of English Language

It seems better to check through the professional editing service.

Author Response

  1. Target on the aerial parts of a Tunisian specimen of Glycyrrhiza foetida(Fabaceae) needs to introduce in detail.

Answer: in the Introduction (lines 54-56) we explain the reason for analyzing the aerial parts of this Tunisian plants. This is a follow-up study of our previous investigation reported in ref 5.

  1. Extraction must follow the previous report(s) with reference(s).

Answer: As suggested, we have shortened part. 2.3 making reference to previous report.

  1. How to prepare the purified compound for bioassay in cells?

Answer: Pure compounds were dissolved in DMSO at different concentrations and added to the cells.  

  1. Assay of mitochondrial activity needs the reference(s) to support. Same as that of glucose uptake in C2C12 cells.

Answer: The two suggested references have been added. (ref. 22 and 24)

5-7.      Data in figures belonged to small size in triplicate only. In vivo data were ignored. Why? Dose-dependent change by each compound is extremely required.

            Answer: We thank the reviewer for pointing out these aspects. Our study is intended to be a preliminary evaluation of the metabolites obtained from this plant source as modulators of interesting targets related to metabolic disease. The reason for not carrying out more experiments, not establishing a dose-dependent curve or, even more, for running in vivo experiments is the same, namely the small amounts of many of the isolated metabolites. In most of the cases, we have isolated 0.5-1.5 mg, which are clearly not enough for doing more experiments than those we have done.

  1. One table indicated the amount of each compound from this plant is helpful.

Answer: We respectfully believe that adding such a table would make the manuscript even longer, with little benefit since all the information about amounts isolated is contained in a single paragraph, namely par. 2.5.

  1. Compound belonged to active in mitochondrial and GLUT stimulation must supported by reliable results.

Answer: We recognize that our results are preliminary, but being statistically significant they are reliable as starting point for future investigations. This has been clearly stated in the last sentence of the Conclusion paragraph.

  1. Phlorizin (25) is known as inhibitor of sodium-glucose transport protein without selectivity for type-1 or type-2. It seems not the same as conclusion.

Answer: Following the reviewer’s suggestion, we have deleted the specification on the type of sodium-glucose transporter.

  1. Criteria of metabolic syndrome needs to make clear. Please check the established reference(s).

     Answer: following the reviewer’s suggestion, we have updated the metabolic syndrome definition in the first sentence of Introduction and changed the ref. 1

  1. Effective dose of each compound was not conducted in clear. Novelty of current report was unknown.

Answer: Please, see answers to points 5—7 and 9

Round 2

Reviewer 1 Report

Comments and Suggestions for Authors

The authors have answered all questions the reviewer had previously. 

Reviewer 3 Report

Comments and Suggestions for Authors

Revision has been performed following the comments.

Comments on the Quality of English Language

It seems better to check through a professional editing service.